# Text Mining and Determinants of Sentiments towards the COVID-19 Vaccine Booster of Twitter Users in Malaysia

**DOI:** 10.3390/healthcare10060994

**Published:** 2022-05-27

**Authors:** Song-Quan Ong, Maisarah Binti Mohamed Pauzi, Keng Hoon Gan

**Affiliations:** 1Institute for Tropical Biology and Conservation, Universiti Malaysia Sabah, Jalan UMS, Kota Kinabalu 88400, Malaysia; 2School of Computer Sciences, Universiti Sains Malaysia, Penang 11800, Malaysia; maisarah.mohd.pauzi@student.usm.my (M.B.M.P.); khgan@usm.my (K.H.G.)

**Keywords:** astrazeneca, Pfizer-BioNTech, RFE, Boruta, sinovac, Twitter, vaccination booster

## Abstract

Vaccination is the primary preventive measure against the COVID-19 infection, and an additional vaccine dosage is crucial to increase the immunity level of the community. However, public bias, as reflected on social media, may have a significant impact on the vaccination program. We aim to investigate the attitudes to the COVID-19 vaccination booster in Malaysia by using sentiment analysis. We retrieved 788 tweets containing COVID-19 vaccine booster keywords and identified the common topics discussed in tweets that related to the booster by using latent Dirichlet allocation (LDA) and performed sentiment analysis to understand the determinants for the sentiments to receiving the vaccination booster in Malaysia. We identified three important LDA topics: (1) type of vaccination booster; (2) effects of vaccination booster; (3) vaccination program operation. The type of vaccination further transformed into attributes of “az”, “pfizer”, “sinovac”, and “mix” for determinants’ assessments. Effect and type of vaccine booster associated stronger than program operation topic for the sentiments, and “pfizer” and “mix” were the strongest determinants of the tweet’s sentiments after the Boruta feature selection and validated from the performance of regression analysis. This study provided a comprehensive workflow to retrieve and identify important healthcare topic from social media.

## 1. Introduction

COVID-19 is an infectious disease caused by severe acute respiratory syndrome coronavirus 2 (SARS-CoV-2) [1] which triggered a global pandemic that had significant negative impacts on individuals, governments, and the global economy [2,3]. As of early January 2022, the worldwide cumulative number of confirmed cases was 334 million and the number of deaths was 5.55 million. In Malaysia, the total number of confirmed cases was 2.81 million, and the number of deaths was 31,818 in early January of 2022 [4]. The incidences and deaths have been rising globally, and with the threat of the new variant—Omicron (variant B.1.1.529) [5]—most countries have been mandating the additional vaccination dose or booster to increase the immunity level in the community. However, the challenges of convincing the public to receive the vaccination remain; without an understanding of public opinion and their concerns about the additional vaccination, the progress of the national vaccination program could be affected.

Social media such as Facebook, Twitter, and Instagram contain a lot of untapped potential for generating non-trivial knowledge and have considerable potential to be used to study the changes in social, public health, and economic arenas. Acquiring information directly from the public through social media is particularly valuable when a timely decision is required, but this is not feasible by traditional communication such as surveys and interviews. Specifically, in Malaysia, social and economic constraints as well as natural disasters (flooding in multiple areas of Malaysia in January 2022 [6]) disrupt the collection of data. With these limitations, text mining could provide an excellent alternative for data collection of public opinion on the additional vaccine in Malaysia. Text mining is an automated technique that uses computational algorithms to extract knowledge and patterns from text, such as the microblog of Twitter [7,8,9]. Text mining, such as information extraction/retrieval, topic modeling, semantic analysis, and associations, is among the machine learning tools that could be developed to generate relevant and timely information from social media platforms. We aim to use text mining to identify the relevant information from large amounts of currently existing unstructured tweets, rather than simply finding words, in the manner of a search engine.

This study is structured as follows. In Section 1.1, more related works are presented and discussed to lead to the research question and the aim of this study and to justify the methodology used in this study. Section 2 details the methodology, from the tools used to retrieve tweets and annotation with sentiments to attributes/feature selection, by using two machine learning approaches validated by logistics regression. For Section 3, we present and discuss the results and, in Section 4, we conclude with the limitations and future directions of the study.

### Related Works

Sentiment analysis is the use of text mining to identify and extract information that can be used as a monitoring tool for the evaluation of projects and assist in decision making [10]. Sentiments on the topic of additional vaccine doses generated from social media content may be the result of listening to the public’s opinion and feelings about the issue. Sentiment analysis can be approached as one or a combination of supervised, semi-supervised, and unsupervised classification tasks. Lexicon and machine learning are popular approaches for sentimental classification [10,11]. Kwok et al. [12] grouped the topics of tweets from Australian Twitter users to obtain their public opinion on the COVID-19 vaccination program and found that attitude misconceptions and complaints about the vaccination played a key role in the sentiments. Ahmed et al. [13] studied the time series of the tweets to obtain the trend and sentiment dynamics on the topics in the COVID-19 pandemic. Ridwan and Hargreaves [14] used Twitter data to understand the public sentiments for the COVID-19 outbreak in Singapore. Nonetheless, topic modeling, extraction, or sentiments analysis on vaccine boosters have yet to be studied. Ansari et al. [15] used COVID-19 vaccine related tweets and conducted sentiment analysis to uncover the latest information on the effect of location and gender on the current vaccination. Aygun et al. [16] used aspect-based sentiment analysis for Twitter users from the USA, UK, Canada, Turkey, France, Germany, Spain, and Italy and used four different aspects (policy, health, media, and other) and four different BERT models (mBERT-base, BioBERT, ClinicalBERT, and BERTurk) to understand peoples’ views about vaccination and types of vaccines. Marcec et al. [17] retrieved all English-language tweets mentioning AstraZeneca/Oxford and Pfizer/BioNTech and conducted sentiment analysis using the AFINN lexicon to calculate the daily average sentiment of tweets to understand the sentiment of tweets on each vaccine.

In this study, we aim to retrieve the tweets that are related to the additional vaccine dose or booster from Malaysia Twitter users, from January to February 2022, due to the rationales of it being the (1) end of the country’s lockdown, (2) peak period for lunar Chinese New Year preparation, and (3) peak period of threat from Omicron. We are particularly interested in the commonly discussed topics on Twitter that are related to vaccine boosters and aim to use latent Dirichlet allocation (LDA) to identify topics from the tweets. We aim to pinpoint the sentiment determinants towards the vaccine booster, and therefore conducted a multinomial regression model to study the association between the topics and their sentiments.

## 2. Materials and Methods

### 2.1. Dataset with Sentiment Extracts

We constructed the dataset by using Rapidminer (Rapidminer Studio Version 9.10) [18]. After obtaining the authorization token to access the tweets, we retrieved the data by using the query of “booster” and/or “vaccine” in the Malaysia region by using the Twitter search operator in Rapidminer. Due to the language of the text being a mixture of Malay and English, we translated the text by leveraging the Google Sheets translation features [19], which use the comment “=GOOGLETRANSLATE” (column that has text, “id”, “en”). Data cleaning was conducted by using regular expression and later using “Replace” operators to replace/clean http tags and RT@ such as “http—https?://[-a-zA-Z0-9+&@#/%?=~_|!:,.;]*[-a-zA-Z0-9+&@#/%=~_|]” and “RT @—RT\s*@[^:]*:\s*[A-Za-z ]+” Quality checking was conducted to remove irrelevant tweets. For the sentiment annotation, we used the “Sentiment Extract” operator in RapidMiner to calculate the sentiment score and annotate the tweets. The SentiWordNet model in the operator applies automatic annotation of all the synsets of WORDNET according to the notions of “positivity”, “negativity”, and “neutrality” [20]. Each synset is associated with three numerical scores, Pos(s), Neg (s), and Obj(s), which indicate how positive, negative, and “objective” (i.e., neutral) the terms contained in the synset are. Different senses of the same term may thus have different opinion-related properties. Each of the three scores ranges in the interval [0.0, 1.0], and their sum is for each synset. This means that a 1.0 synset may have nonzero scores for all the three categories, which would indicate that the corresponding terms have, in the sense indicated by the synset, each of the three opinion-related properties to a certain degree. The process flow of dataset construction is illustrated in Figure 1.

### 2.2. Topic Modelling with LDA

Latent Dirichlet allocation (LDA) is an unsupervised machine learning method that allows observations such as words or documents in a corpus to be explained by latent groups, such as topics [21]. We used the RapidMiner “Extract Topic from Text” operator to implement LDA. The implementation of LDA uses the ParallelTopicModel of the Mallet library and Gibbs Sampling for the application of the model [22]. We deployed 1000 iterations on the top five topics, and only the top 100 words with the highest weights were visualized using a word cloud for each topic. Larger font size and a higher level of opacity were used to indicate words with higher beta values. In each topic, the top 20 tweets—except those from news sources—with the highest weight, which were also larger than those of the other two topics for each tweet, were reported.

### 2.3. Feature Selection

One of the goals of this study was to assess the topics related to vaccine boosters as the determinants and their association to the sentiment polarity. We derived the most discussed topics from the LDA and aimed to use them and their attributes to train a multinomial regression model to study the association between determinants’ and sentiments’ polarity. We implemented two feature selection methods: Boruta and Recursive Feature Elimination (RFE) [23], to select the features from attributes derived from LDA. Figure 2 illustrates the process of feature assessment and representative feature identification before the association study. Feature assessment was performed using R version 4.1.1 with the Boruta package for Boruta and the caret package for RFE.

### 2.4. Sentiment Determinant Association Study

To study the association between potential attributes and their sentiments, we used a multinomial regression model that can discriminate three sentiments—positive, neutral, and negative. As we obtained three top topics from LDA which were later derived into a total of five attribute groups, we built five multiple logistics regression models by using the inputs of these five attribute groups, respectively. Evaluation matrices of accuracy, sensitivity, and specificity were used to compare the performance of the models. The data splitting and model development were accomplished using R packages of tidyverse, nnet, caret, and dplyr [24]. Figure 3 shows the process of data splitting and model build-up by using two groups of input data.

## 3. Results

### 3.1. Sentiment Polarity Distribution

In general, nearly half of the sentiments of all tweets expressed a positive public opinion about the vaccine booster, around one-third were negative, and less than 20% were neutral (Figure 4).

To be specific, we evaluated the sentiments of the topic statistically by referring to its probability. As can be seen from Figure 5, the program operation had a higher probability of positive sentiment, which indicated positive opinion from the public and, therefore, the success of the country vaccination program. On the other hand, three topics consisted of higher negative sentiments, being “effect”, “sino”, and “mix”. Most of the “effect” related tweets were related to the effect of getting additional vaccines such as fever, pain, death, etc., whereas “sino” was related to the concern of getting Sinovac as the additional vaccine booster, and “mix” was related to content where having more than one vaccine was mentioned.

### 3.2. Topic Modelling with LDA

LDA has been used in topic modeling of public opinions on COVID-19 vaccination since the beginning of the pandemic [11]; nonetheless, it had yet to be used for the determination of public opinion on the additional vaccine booster. We first analyzed the preprocessed tweets by visualizing the word tokens with a count of >100 in the corpus, as shown in the word cloud in Figure 6. The larger the word font size in the cloud, the higher the number of counts in the corpus. The top seven high-frequency words were “Pfizer”, “Appointment”, “Fever”, “az”, “PPV”, “My_Sejahtera”, and “effects”. Based on the descriptive statistics of word counts, topics such as the type of vaccination booster, effects of getting the additional vaccination, and the national vaccination program were the major discussion topics among Malaysian citizens.

We used the LDA to discover the topics discussed by Malaysian Twitter users. For this study, three topics were pre-assigned based on the word cloud generated in Figure 6, and the results of the words’ allocation are summarized in Table 1.

### 3.3. Feature Selection

We aimed to study the relevance and importance of the attributes by utilizing the algorithm of feature selection methods. We derived six attributes from the top three topics modeled from LDA, and Figure 7 and Figure 8 show the outcome of the RFE and Boruta result. Figure 9 summarizes the result of feature selection and Figure 10 the process to assess the features to predict the sentiment polarity of tweets.

### 3.4. Multinomial Logistics Regression Model

We aim to pinpoint the sentiment determinants related to the vaccine booster by studying the association between all possible and representative attributes with the sentiment’s polarity. Our result shows the model that has two representative attributes—“pfizer” and “mix” was the top-performing model, which achieved 0.7143 accuracies, sensitivity of 0.333, and specificity of 0.6667. Figure 11 demonstrates the model performance for all and representative attributes, and Figure 12 shows the confusion matrix.

## 4. Discussion

For the sentiment polarity distribution, our result revealed a high probability of the topic of program operation and concern over getting a mixture of different brands of vaccines as the additional vaccine booster. This was a similar result as Marcec et al. [17], who retrieved tweets mentioning AstraZeneca/Oxford and Pfizer/BioNTech and found that a mixture of them showed negative polarity of tweets. For LDA topic modelling, since LDA assumes each topic is made of a bag of words with certain probabilities, and the algorithm learns the word and topic distribution underlying the corpus, this study considered the words that were allocated under each topic to assign meaning to the bag of words under each topic. Our results paralleled those of Ridwan and Hargreaves [14], who used Twitter data to understand the public sentiments and topics for the COVID-19 outbreak in Singapore, for which the topics of effect and the program operation generate most of the sentiments.

High dimensional data pose a great challenge for computational techniques, and reducing the number of features has four main advantages: it decreases computational costs, mitigates the possibility of overfitting due to high inter-variable correlations, secures model performance, and allows for an easier clinical interpretation of the model. We aimed to study the relevance and importance of the attributes by utilizing the algorithm of feature selection methods. This study retrieved 788 tweets from Malaysia Twitter users from January to February 2022. We derived six attributes from the top three topics modeled from LDA. We further assessed the attributes by using two feature selection methods: RFE (Figure 7) rigorously filtered all the variables and found five attributes highly related to the target outcome, and Boruta further selected the two most important attributes (Figure 8) as the representative determinants. Feature selection is a standard procedure to filter or select the important attributes that determine the sentiments of the public [25], and we further pinpointed the sentiment determinants related to the vaccine booster by using multivariance logistics regression. This study’s procedures are standard across other infectious diseases [15,16,25], and the results of our model showing two representative attributes—“pfizer” and “mix”—were supported by Ahmed et al. [13], Aygun et al. [16], Marcec et al. [17], etc., in that the brand of vaccine played a crucial role when the public plan was to administer an additional vaccine booster.

## 5. Conclusions

Our findings indicate that the Malaysian public possessed varying concerns toward the COVID-19 additional vaccine booster. In general, these can be grouped into (1) type of vaccine booster, (2) effect of vaccination, and (3) vaccination program operation. By referring to the individual topic sentiments, the public had higher probabilities of positive sentiments towards the country vaccination program, but showed higher negative sentiments towards Sinovac and the mixing of vaccine boosters. We confirmed the results by both features selection and the multinomial regression model, in which “pfizer” and “mix” were the most important determinants for the topic of the vaccine booster. The method demonstrated in this study was feasible and effective for authorities to be mindful of public opinion through text mining of social media in a country where the traditional methods of data collection are hampered. 

Due to the specific duration of retrieving the information from Twitter, we were limited by the available number of tweets; however, our results are encouraging and should be explored in a larger cohort of the population to obtain the sentiments related to getting vaccination boosters. Further work is needed to validate the sentiments of the public towards the vaccination booster via formal survey.

## Figures and Tables

**Figure 1 healthcare-10-00994-f001:**
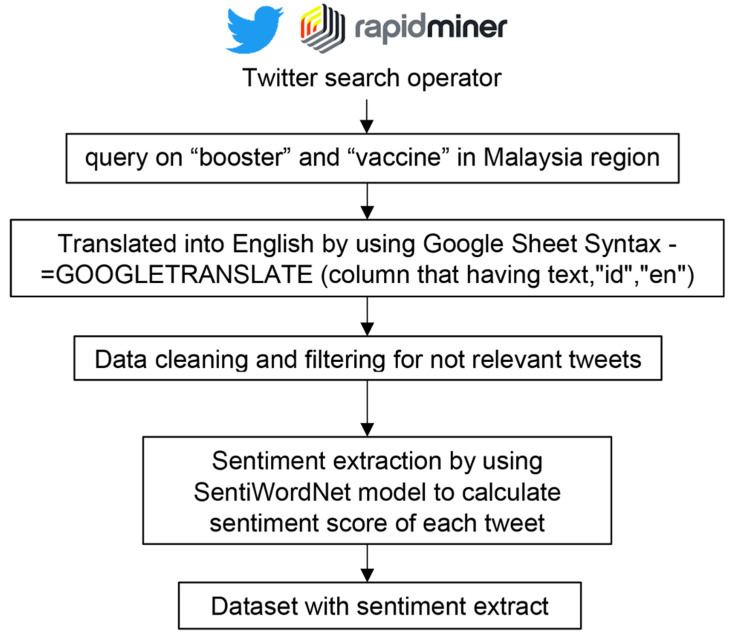
Process flow of dataset construction.

**Figure 2 healthcare-10-00994-f002:**
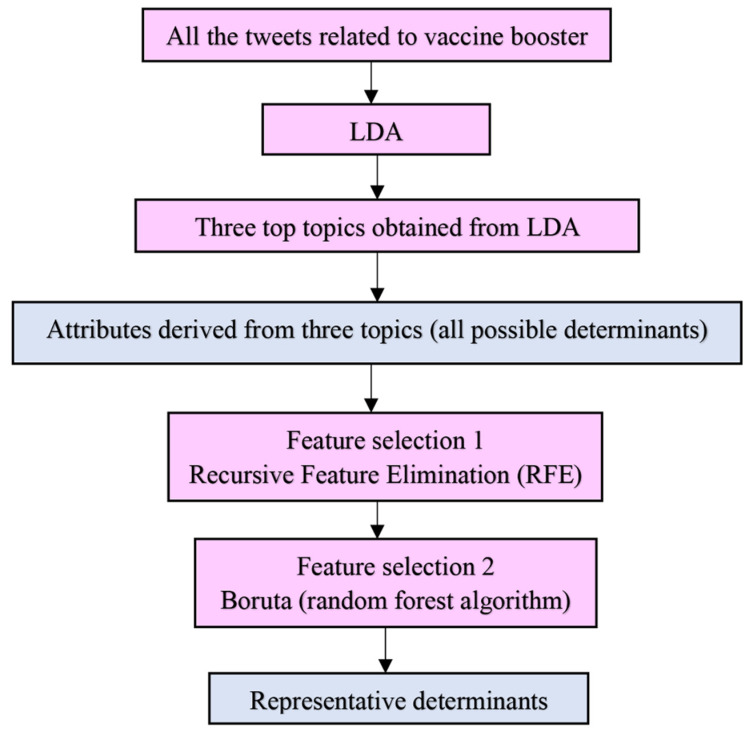
Process of feature assessment.

**Figure 3 healthcare-10-00994-f003:**
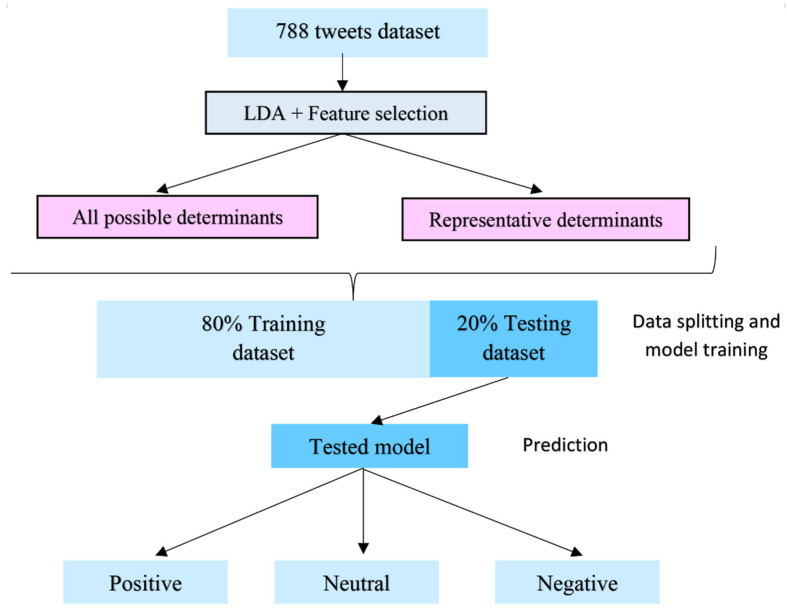
The process of multinomial regression model build-up with two different groups of features as input data.

**Figure 4 healthcare-10-00994-f004:**
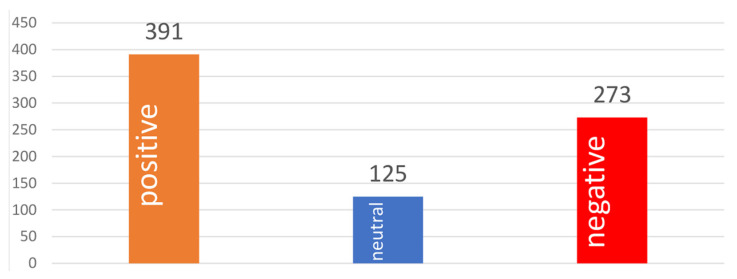
Sentiment polarity distribution of the tweets.

**Figure 5 healthcare-10-00994-f005:**
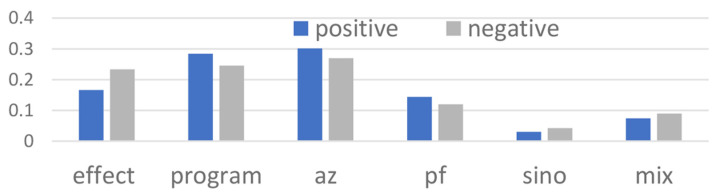
Topic sentiments’ probabilities.

**Figure 6 healthcare-10-00994-f006:**
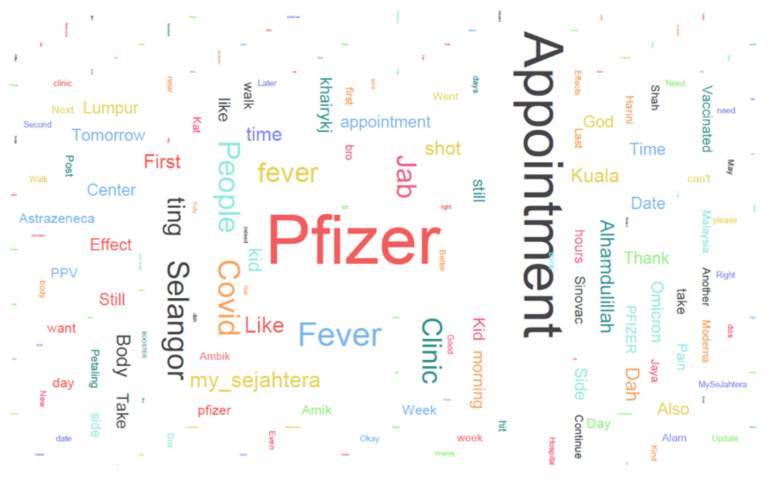
Word cloud.

**Figure 7 healthcare-10-00994-f007:**
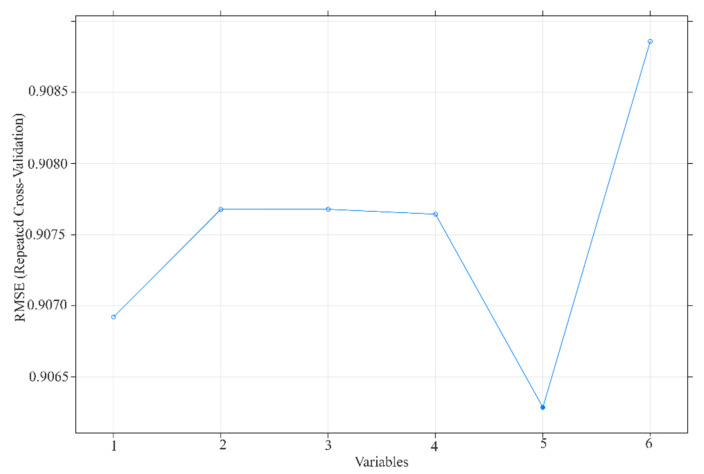
RFE result—ranking of variables based on the model Root Mean Square Error (RMSE). X-axes represent the subset sizes of variables; Y-axes represent the RMSE of the residuals, which indicate the prediction strength of the variables.

**Figure 8 healthcare-10-00994-f008:**
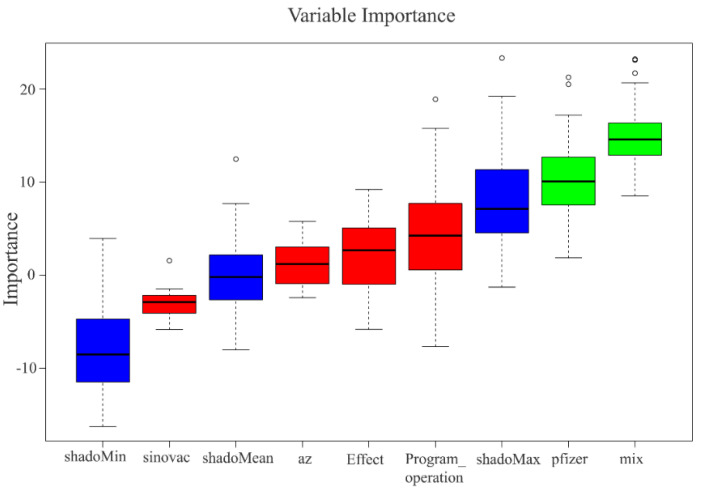
Boruta result plot for six possible determinants. Blue boxplots represent the minimal, average, and maximum Z-score of a shadow attribute. Red and green boxplots represent Z-scores of respectively rejected and confirmed attributes.

**Figure 9 healthcare-10-00994-f009:**
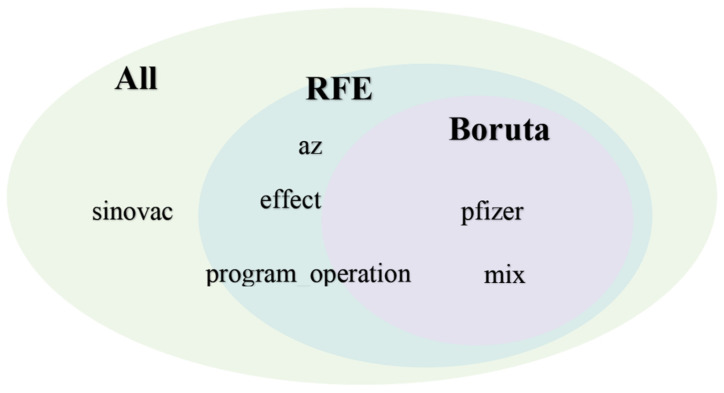
Venn diagram illustrating all, RFE, and Boruta determinants.

**Figure 10 healthcare-10-00994-f010:**
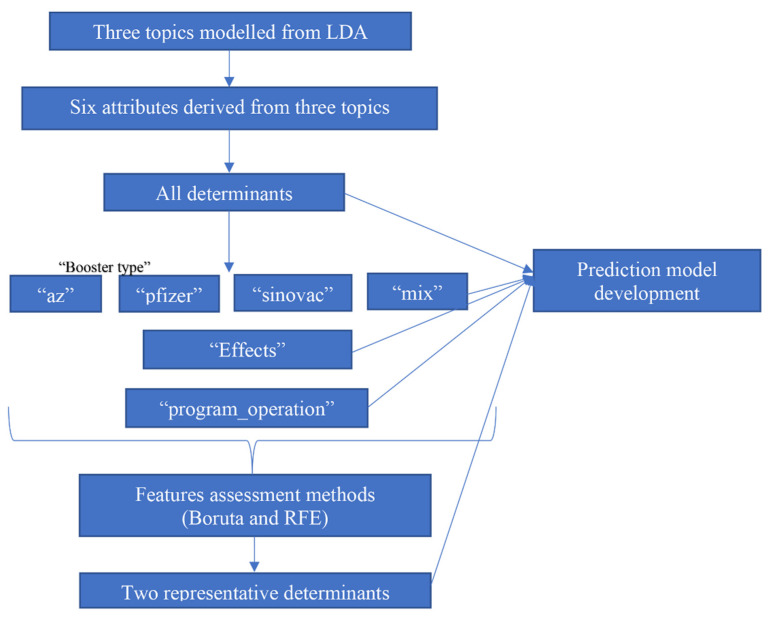
Feature assessment process and numbers of features obtained from each method.

**Figure 11 healthcare-10-00994-f011:**
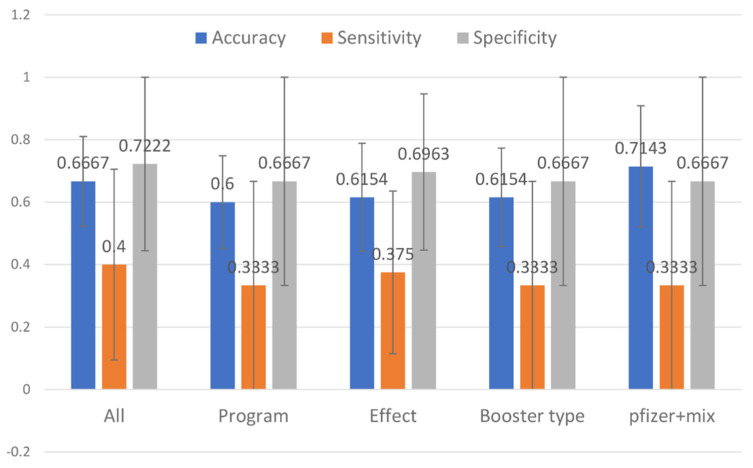
Multinomial regression model performance using all and representative attributes.

**Figure 12 healthcare-10-00994-f012:**
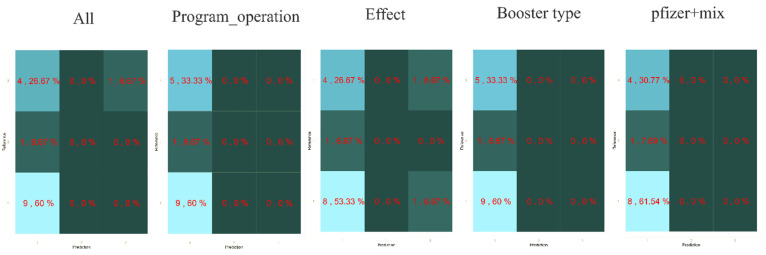
Confusion matrix.

**Table 1 healthcare-10-00994-t001:** Topics modeled from the tweets with LDA for the topic related to vaccine booster in Malaysia.

Topic 1 (Type of Booster)	Topic 2 (Effect of Vaccination)	Topic 3 (Vaccination Program)
Pfizer	die	Khairykj
astrazeneca	Fever	Appointment
Sinovac	Effect	Selangor
	moody	Clinic
	Stress	appt
	symptoms	Walk-in
	tido	mysejahtera
	Pain	
	Side	

## Data Availability

The data presented in this study are available on request from the corresponding author.

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
