# Peer review of "Text Mining and Determinants of Sentiments towards the COVID-19 Vaccine Booster of Twitter Users in Malaysia"

_healthcare, 2022, doi:10.3390/healthcare10060994_

Round 1
Reviewer 1 Report
The article tries to draw conclusions based on a relatively small number (less than 1000) of tweets. In my opinion, no reliable conclusions can be drawn from such a small sample, so the starting point cannot be considered good. This could still be changed by increasing the amount of data examined. The main shortcoming of the article is that it does not provide a new method, new essential conclusions, as it only processes data in one process using services provided by known programs, so it cannot be considered as more than simple data processing using software. For this reason, it cannot be evaluated scientifically. It can be presented at a smaller conference as a case study, but it is not enough to appear in a journal.
The literature review is also scarce and inadequate. Compared to the articles listed, many better, fresher, more thorough articles could have been found.
For this reason, I do not recommend the publication of this article.
Author Response
Response to Reviewer 1 Comments
Point 1: The article tries to draw conclusions based on a relatively small number (less than 1000) of tweets. In my opinion, no reliable conclusions can be drawn from such a small sample, so the starting point cannot be considered good. This could still be changed by increasing the amount of data examined. The main shortcoming of the article is that it does not provide a new method, new essential conclusions, as it only processes data in one process using services provided by known programs, so it cannot be considered as more than simple data processing using software. For this reason, it cannot be evaluated scientifically. It can be presented at a smaller conference as a case study, but it is not enough to appear in a journal.
The literature review is also scarce and inadequate. Compared to the articles listed, many better, fresher, more thorough articles could have been found.
For this reason, I do not recommend the publication of this article
Response 1:
Thank you very much for the comments. Introduction has been rewrite and added a section of "related work" to included more related research that able to lead to the research question and objective. Limitation of the number of tweets and justification, and future direction of research also added in conclusion. Although the recommendation is least favorite to us, we are still pleased to enclosed the revised manuscript with tracking comments in the attachment. Thanks again for the review.
Reviewer 2 Report
In a country, such as Malaysia, where there are impediments to obtaining traditional information about public sentiment related to COVID-19 boosters, these authors argue and demonstrate that text mining Twitter is a reasonable and effective method to glean this information.
The strengths of this paper are that the introduction is appropriate and well-referenced, the materials and methods are clearly explained and the figures are plentiful, relevant and well situated. The weaknesses are that there are references missing in the materials and methods section, some of the figures are illegible because of their size and/or their colour, Table 1 is missing, the conclusion goes beyond the information provided in the paper, and the references were not done in accordance with MDPI style.
Line by line suggested edits
13 Change “machine learning” to “text mining” as machine learning is not used anywhere in the body of the paper.
24 Eliminate LASSON from the keywords as it is not found in the abstract nor in the body of the paper. Add Sinovac, vaccination booster, Twitter, social media and text mining to the list of keywords.
43 Change “feasible to be covered by“ to “feasible by”.
44-45 Change “in Malaysia that consists of constraints of social, economic, and natural disasters (flood in multiple areas of Malaysia in Jan 2022 [6]) that” to “in Malaysia, social and economic constraints as well as natural disasters (flood in multiple areas of Malaysia in Jan 2022 [6])”.
79 Please provide a reference for Rapidminer.
83 Please provide a reference for Google sheets translation features.
115 Please provide a reference for Boruta.
115-116 Please provide a reference for Recursive Feature Elimination (RFE)
129 Please provide a reference for R packages of tidyverse, nnet, caret, and dplyr.
142 Change “was having a” to “had”.
153 Change “there was yet to be done” to “it was yet to be done”.
160 Change “were major” to “were the major”.
163 Change “Latent Dirichlet Allocation (LDA)” to “LDA”.
165-166 Table 1 is missing.
175-177 Given that in the previous sentences the authors have argued that there are three advantages to reducing the number of attributes, the sentence, “Although the total attributes we derived from this study are only six, we aim to study the relevancy and importance of the attributes by utilizing the algorithm of feature selection methods.”, is inappropriate because it is written as if there is a problem with having only six attributes. As well, the relevant information in this sentence is repeated in the sentences to follow. Please eliminate this sentence as a result.
179 This text states that the method depicted in Figure 7 is RFE yet the heading provided to Figure 7 in 186-188 does not mention RFE. Please rewrite the heading for Figure 7 in lines 186-188 to mention RFE.
182 Change “Figure 9 and 10 summarises” to “Figure 9 and 10 summarize”.
Figure 7 is too small. The information on the figure is illegible.
Figure 8 is too small. The information on the figure is illegible.
194 Change “Figure 9. Venn diagram to illustrates all and representative determinants.” to “Figure 9. Venn diagram illustrating all, RFE and Boruta determinants.”
Figure 10 Please enlarge Figure 10 so the font within the boxes is the same size as the text font in the body of the paper.
Figure 12 The information on this figure is illegible. Please lighten the colour of the blues used so that the information in the individual matrices can be read.
212 Change “was having higher” to “had higher”.
216-218 Given that there was no discussion on education by governments in the paper, this type of conclusion cannot be drawn. What can be stated is that, regarding boosters for COVID-19, it has been found reasonable and effective for governments to be mindful of public opinion through text mining of social media in a country like Malaysia where the collection of data through traditional means is hampered.
The Data Availability Statement: has been left out. Please include this statement.
Please redo the References as per the examples on the Healthcare template to conform to MDPI style.
Author Response
Response to Reviewer 2 Comments
A huge thank you to reviewer for the valuable advice and comments.
Point 1: The weaknesses are that there are references missing in the materials and methods section, some of the figures are illegible because of their size and/or their colour, Table 1 is missing, the conclusion goes beyond the information provided in the paper, and the references were not done in accordance with MDPI style.
Response 1:
Thank you very much for the comments. The references were added in the materials and methods section accordingly. Figures were re-constructed into high resolution Table 1 was added, conclusion was rewrite and added with limitation and future direction of the research The references were redo according to MDPI style.
Point 2: Change “machine learning” to “text mining” as machine learning is not used anywhere in the body of the paper.
Response 2: Together with comments of another reviewer, the sentence has been revised into: "We aim to investigate the attitudes to COVID-19 vaccination booster in Malaysia by using sentiment "
Point 3: Eliminate LASSON from the keywords as it is not found in the abstract nor in the body of the paper. Add Sinovac, vaccination booster, Twitter, social media and text mining to the list of keywords.
Response 3: LASSON has been removed, and Sinovac, vaccination booster, Twitter have been added in the keywords
Point 4: Change “feasible to be covered by“ to “feasible by”.
Response 4: “feasible to be covered by“ has been changed to “feasible by”
Point 5: Change “in Malaysia that consists of constraints of social, economic, and natural disasters (flood in multiple areas of Malaysia in Jan 2022 [6]) that” to “in Malaysia, social and economic constraints as well as natural disasters (flood in multiple areas of Malaysia in Jan 2022 [6])”.
Response 5: The sentence has been replaced.
Point 6: Please provide a reference for Rapidminer.
Response 6: A reference [18] Vyas, V.; Uma, V. An extensive study of sentiment analysis tools and binary classification of tweets using rapid miner. Procedia Comput. Sci., 2018, 125, 329-335. has been added for Rapidmine
Point 7: Please provide a reference for Google sheets translation features.
Response 7: A reference [19] Ali, A. Using Google Docs to Enhance Students’ Collaborative Translation and Engagement. J. Inf. Technol. Educ.:Res., 2021, 20, 503-528. was added for Google sheets translation features
Point 8: Please provide a reference for Boruta.
Response 8: A reference [23] Nahar, N.; Ara, F.; Neloy, M.; Istiek, A.; Biswas, A.; Hossain, M. S.; Andersson, K. Feature Selection Based Machine Learning to Improve Prediction of Parkinson Disease. In International Conference on Brain Informatics, 2021, pp. 496-508). Springer, Cham. Has been added for Boruta and RFE
Point 9: Please provide a reference for Recursive Feature Elimination (RFE)
Response 9: A reference [23] Nahar, N.; Ara, F.; Neloy, M.; Istiek, A.; Biswas, A.; Hossain, M. S.; Andersson, K. Feature Selection Based Machine Learning to Improve Prediction of Parkinson Disease. In International Conference on Brain Informatics, 2021, pp. 496-508). Springer, Cham. Has been added for Boruta and RFE
Point 10: Please provide a reference for R packages of tidyverse, nnet, caret, and dplyr.
Response 10: A reference [24] Wickham, H., Averick, M., Bryan, J., Chang, W., McGowan, L. D. A., François, R., ... & Yutani, H. Welcome to the Tidyverse. J. Open Source Softw., 2019, 4(43), 1686. has been added for the R packages,
Point 11: Change “was having a” to “had”.
Response 11: The term has been revised
Point 12: Change “there was yet to be done” to “it was yet to be done”.
Response 12: The term has been revised
Point 13: Change “were major” to “were the major”.
Response 13: The term has been revised
Point 14: Change “Latent Dirichlet Allocation (LDA)” to “LDA”.
Response 14: The term has been revised
Point 15: Table 1 is missing.
Response 15: We are sorry for the mistake, Table 1 is added.
Point 16: Given that in the previous sentences the authors have argued that there are three advantages to reducing the number of attributes, the sentence, “Although the total attributes we derived from this study are only six, we aim to study the relevancy and importance of the attributes by utilizing the algorithm of feature selection methods.”, is inappropriate because it is written as if there is a problem with having only six attributes. As well, the relevant information in this sentence is repeated in the sentences to follow. Please eliminate this sentence as a result.
Response 16: The sentence has been removed
Point 17: This text states that the method depicted in Figure 7 is RFE yet the heading provided to Figure 7 in 186-188 does not mention RFE. Please rewrite the heading for Figure 7 in lines 186-188 to mention RFE.
Response 17: Heading for Figure 7 has been rewrite.
Point 18: Change “Figure 9 and 10 summarises” to “Figure 9 and 10 summarize”.
Response 18: The term has been revised
Point 19: Figure 7 is too small. The information on the figure is illegible.
Response 19: The font size of figure has been enlarged
Point 20: Figure 8 is too small. The information on the figure is illegible.
Response 20: The font size of figure has been enlarged
Point 21: Change “Figure 9. Venn diagram to illustrates all and representative determinants.” to “Figure 9. Venn diagram illustrating all, RFE and Boruta determinants.”
Figure 10 Please enlarge Figure 10 so the font within the boxes is the same size as the text font in the body of the paper.
Figure 12 The information on this figure is illegible. Please lighten the colour of the blues used so that the information in the individual matrices can be read
Response 21: Figure 9 - Heading has been changed
Figure 10 - the figure has been enlarge so that the font size is the same as the text font size
Figure 12 - the colour has been lighten and increase the contrast to make the digit become visible
Point 22: Change “was having higher” to “had higher”
Response 22: The term has been revised
Point 23: Given that there was no discussion on education by governments in the paper, this type of conclusion cannot be drawn. What can be stated is that, regarding boosters for COVID-19, it has been found reasonable and effective for governments to be mindful of public opinion through text mining of social media in a country like Malaysia where the collection of data through traditional means is hampered.
Response 23: The sentences has been replaced with "The method demonstrated in this study was feasible and effective for authority to be mindful of public opinion through text mining of social media in a country where the traditional methods of data collection is hampered.
Point 24: The Data Availability Statement: has been left out. Please include this statement.
Response 24: The Data Availability Statement has been added.

Reviewer 3 Report
In the paper presented here, the authors focused on analyzing sentiment related to covid 19 vaccination booster. This topic is very current and important.
I suggest making some improvements:
- The authors formulated the following purpose of the article (line 12) “We aim to understand the topics related to vaccination booster and their sentiments on Twitter by 12 using machine learning”. In my opinion, it is not appropriate for the research results presented. I suggest replacing it with, for example, „Investigating attitudes to covid 19 vaccination booster in Malaysia region using sentiment analysis”.
- At the end of the Introduction section, I suggest adding a description of the structure of the article, i.e., what is included in each section.
- I propose to add a Related Works section that will present findings from other papers on the use of sentiment analysis to determine attitudes toward Covid 19 vaccines. Many such papers have appeared in the last year. You can also refer to the results from these papers in the discussion of the studies. I suggest you read the following literature:
- Ansari MTJ, Khan NA. Worldwide COVID-19 Vaccines Sentiment Analysis Through Twitter Content. Electron J Gen Med. 2021;18(6):em329. https://doi.org/10.29333/ejgm/11316
- Kazi Nabiul Alam, Md Shakib Khan, Abdur Rab Dhruba, Mohammad Monirujjaman Khan, Jehad F. Al-Amri, Mehedi Masud, Majdi Rawashdeh, "Deep Learning-Based Sentiment Analysis of COVID-19 Vaccination Responses from Twitter Data", Computational and Mathematical Methods in Medicine, vol. 2021, Article ID 4321131, 15 pages, 2021. https://doi.org/10.1155/2021/4321131
- Aygun I, Kaya B, Kaya M. Aspect Based Twitter Sentiment Analysis on Vaccination and Vaccine Types in COVID-19 Pandemic with Deep Learning. IEEE J Biomed Health Inform. 2021 Dec 7;PP. doi: 10.1109/JBHI.2021.3133103. Epub ahead of print. PMID: 34874877.
- Marcec R, Likic R, Using Twitter for sentiment analysis towards AstraZeneca/Oxford, Pfizer/BioNTech and Moderna COVID-19 vaccines, Postgraduate Medical Journal Published Online First: 09 August 2021. doi: 10.1136/postgradmedj-2021-140685
- In the Conlusions section, I propose to add further research directions and limitations to the research conducted. The first limitation should be the number of tweets analyzed and the time period over which the analysis was conducted.
Author Response
Response to Reviewer 3 Comments
Point 1: The authors formulated the following purpose of the article (line 12) “We aim to understand the topics related to vaccination booster and their sentiments on Twitter by 12 using machine learning”. In my opinion, it is not appropriate for the research results presented. I suggest replacing it with, for example, „Investigating attitudes to covid 19 vaccination booster in Malaysia region using sentiment analysis”
Response 1:
The sentence has been revised into:"We aim to investigate the attitudes to COVID-19 vaccination booster in Malaysia by using sentiment "
Point 2: At the end of the Introduction section, I suggest adding a description of the structure of the article, i.e., what is included in each section.
Response 2:
A paragraph to describe the structure of articles has been added from line 58-64. Thank you very much for the advise.
Point 3: I propose to add a Related Works section that will present findings from other papers on the use of sentiment analysis to determine attitudes toward Covid 19 vaccines. Many such papers have appeared in the last year. You can also refer to the results from these papers in the discussion of the studies. I suggest you read the following literature:
Ansari MTJ, Khan NA. Worldwide COVID-19 Vaccines Sentiment Analysis Through Twitter Content. Electron J Gen Med. 2021;18(6):em329. https://doi.org/10.29333/ejgm/11316
Kazi Nabiul Alam, Md Shakib Khan, Abdur Rab Dhruba, Mohammad Monirujjaman Khan, Jehad F. Al-Amri, Mehedi Masud, Majdi Rawashdeh, "Deep Learning-Based Sentiment Analysis of COVID-19 Vaccination Responses from Twitter Data", Computational and Mathematical Methods in Medicine, vol. 2021, Article ID 4321131, 15 pages, 2021. https://doi.org/10.1155/2021/4321131
Aygun I, Kaya B, Kaya M. Aspect Based Twitter Sentiment Analysis on Vaccination and Vaccine Types in COVID-19 Pandemic with Deep Learning. IEEE J Biomed Health Inform. 2021 Dec 7;PP. doi: 10.1109/JBHI.2021.3133103. Epub ahead of print. PMID: 34874877.
Marcec R, Likic R, Using Twitter for sentiment analysis towards AstraZeneca/Oxford, Pfizer/BioNTech and Moderna COVID-19 vaccines, Postgraduate Medical Journal Published Online First: 09 August 2021. doi: 10.1136/postgradmedj-2021-140685
Response 3:
Section 1.1 Related work has been added. Thank you very much for the advise.
Point 4: In the Conlusions section, I propose to add further research directions and limitations to the research conducted. The first limitation should be the number of tweets analyzed and the time period over which the analysis was conducted
Response 4:
Limitation and future work have been added in the conclusion.
Reviewer 4 Report
- Lack of an appropriate theoretical introduction indicating the importance of social media in shaping attitudes, in the discussed case in the context of vaccination;
- A small sample of analyzed Tweets from a very short period of time means that drawing conclusions based on the obtained results may be burdened with randomness;
- Section "discussion" is not presented, section "conclusions" is to short and does not present any concluding remarks
- minor spelling and language mistakes
Author Response
Response to Reviewer 4 Comments
Point 1: Lack of an appropriate theoretical introduction indicating the importance of social media in shaping attitudes, in the discussed case in the context of vaccination;
Response 1:
Introduction has been rewrite and added a section of "related work".
Point 2: Section "discussion" is not presented, section "conclusions" is to short and does not present any concluding remarks
Response 2:
Discussion is combined with the section of result to facilitate the content for discussion. Conclusion has been revised to include limitation and future direction of research
Point 3 A small sample of analyzed Tweets from a very short period of time means that drawing conclusions based on the obtained results may be burdened with randomness;
Response 3:
Section 1.1 Related work has been added. Thank you very much for the advise.
Point 4: In the Conlusions section, I propose to add further research directions and limitations to the research conducted. The first limitation should be the number of tweets analyzed and the time period over which the analysis was conducted
Response 4:
Number of tweets and duration have been mentioned in the conclusion as the limitation of the study
Point 5: minor spelling and language mistakes
Response 5:
The manuscript has been edited grammatically by native English-speaking colleague
Round 2
Reviewer 1 Report
Unfortunately, there has been no substantive change in the content of this article. So my opinion hasn’t changed either, it’s very little to appear in a journal. This may be enough for a conference article. I do not recommend the publication of this article.
Author Response
Thank you very much for your comments.

Reviewer 3 Report
The authors have made all the suggested modifications in the paper. I accept the paper in present form.
Author Response
Thank you very much for your valuable comments and feedbacks.

Reviewer 4 Report
The introduced changes do not cover the scope of the comments presented in the first review
Author Response
Point 2: Section "discussion" is not presented, section "conclusions" is to short and does not present any concluding remarks
Section of "discussion" has been added to discuss the result of the study.
